# A Facile and Cost-Effective Method to Prepare Biodegradable Poly(ester urethane)s with Ordered Aliphatic Hard-Segments for Promising Medical Application as Long-Term Implants

**DOI:** 10.3390/polym14091674

**Published:** 2022-04-20

**Authors:** Jingjing Bi, Yifan Liu, Jiaxu Liu

**Affiliations:** College of Chemistry, Chemical Engineering and Materials Science, Shandong Normal University, Jinan 250014, China; jingjing_bi123@163.com (J.B.); ljx13130376900@163.com (J.L.)

**Keywords:** poly(ester urethane)s, cost-effective, bio-degradable, ordered hard segments

## Abstract

The article below describes a simple methodology to prepare cost-effective biodegradable poly(ester urethane)s (PEUs) with ordered hard segments (OHS) for medical application as long-term implants. A low-cost diurethane diol (1,4-butanediol-hexanediisocyanate-1,4-butanediol, BHB) was first designed and synthesized. Consequently, the BHB was employed as a chain extender to react with NCO-terminated poly(*ε*-caprolactone) to obtain PEUs. The molecular structural formats for BHB and PEUs were defined through NMR, FT-IR, and MS together with GPC, while the influence of OHS content on physical/chemical features for casted PEU films was investigated. The introduction of OHS could contribute to forming denser hydrogen-bonds, and consequently produce a compact network structure, resulting in great tensile capacity, low water absorption, and slow hydrolytic degradation rate by PEU films. PEU-2.0 films, which possessed the highest OHS content within PEUs, exhibited 40.6 MPa tensile strength together with 477% elongation at break, 4.3 wt % equilibrium water absorption and only 29.5% weight loss post-12 months’ degradation. In addition, cytotoxicity analysis of film extracts indicated that the cell viability of all PEUs containing OHS exceeded 75%, indicating good cytocompatibility. Due to outstanding tensile features, high biostability, nontoxic and absorbable degradation products and acceptable cytocompatibility, the cost-effective materials exhibited promising applications in the field of long-term implants.

## 1. Introduction

Polyurethane (PU), a type of polymer containing carbamate group, can be designed to have a large degree of freedom, and its physical/chemical features could become regulated by modifying the class and proportion of soft/hard regions [1,2]. Besides such adjustable tensile features, the micro-phase segregation stemming from incompatibility of soft/hard regions endows PU with adequate biocompatibility. These advantages make it widely employed within medicine, such as for artificial skin/tissues/organs, surgical sutures, vascular grafts, and drug carriers [3,4,5,6,7]. 

In an effort to diminish the influence of biological materials over human organs, medical materials employed in the body are required to have biological stability in the early stage of use and can be gradually degraded and absorbed in the later stage [8]. Therefore, both soft and hard segments of PU are expected to degrade into non-toxic and absorbable small molecules. Polyesters, especially poly(*ε*-caprolactone) (PCL) and polylactide, are susceptible to hydrolysis and are typically employed as soft segments to prepare biodegradable PUs [9,10,11,12]. However, the hard segments of PUs are based on diisocyanate, which can produce diamines as degradation products. One method for obtaining excellent tensile features, commercial medical-grade PUs for long-term medical implementations, including Biospan^TM^, Elasthane^TM,^ and Biomer^®^ [13,14], is derived from aromatic diisocyanate. The π-π actions between adjacent aromatic rings could generate inter-chain actions and enhance tensile stress [15]. Aromatic diamines, being reported to be carcinogenic, are generated and discharged throughout breakdown processes [16], even though accumulated aromatic diamine levels within individuals are strongly debated [17,18]. Certainly, PUs with hard segments developed through aliphatic diisocyanate can mitigate the risk stemming from degradation product/s of non-toxic biological dimine [19,20], while poor tensile features limit their application as long-term implants.

Ordered hard segments (OHS) are found to be helpful for acquiring excellent tensile features. Through the adoption of macrodiisocyanate as a chain extender, Spaans et al. [21,22] prepared PUs containing longer OHS with such material revealed as possessing outstanding tensile features. An aliphatic diurethane diisocyanate with an ordered structure (hexanediisocyanate-1,4-butanediol-hexanediisocyanate, HBH) was successfully synthesized by the Hou group, employed as a chain extender to prepare medical biodegradable poly(ester-urethane)s through direct-chain extending with polyester diol [23,24,25]. The corresponding film materials possessed equivalent (or even elevated) tensile features in comparison to commercial clinical PUs of Biospan^TM^ and Chronoflex^TM^ AR. In addition, the degradation products from the hard segments were the less harmful aliphatic diamines that could be absorbed into the body [19,20]. However, storage of such macrodiisocyanate was difficult due to the active terminal -NCO groups. Furthermore, excess expensive hexanediisocyanate was employed to synthesize HBH, resulting in the high cost of HBH. These two shortcomings limit its further application. 

This investigation provided a simple method to prepare the cost-effective biodegradable poly(ester-urethane)s (PEUs) containing OHS, which were expected to have excellent tensile features and slow degradation rate for further application as long-term implants. A low-cost diurethane diol (1,4-butanediol-hexanediisocyanate-1,4-butanediol, BHB) was first designed and synthesized, and consequently, PEUs were generated through a coupling reaction for BHB and NCO-terminated PCL. The chemistry-based compositions for BHB and PEUs were defined, while the impact of OHS content over physicochemical features for casted PEU films was investigated. Furthermore, film cytocompatibility was elucidated through cytotoxicity assessments.

## 2. Materials and Methods 

### 2.1. Materials

PCL (M_n_ = 1000 g/mol) and 1,4-butanediol (BDO, 99%) were procured through Aladdin Reagent Co.™ (Shanghai, China) and vacuum-dehydrated (five hours/100 °C) before utilization. Hexanediisocyanate (HDI, 99.6%) and stannous octoate (Sn(Oct)_2_, >95%) were procured through Sigma-Aldrich (Shanghai) Trading Co., Ltd., (Shanghai, China). N,N-Dimethylformamide (DMF), trichloromethane (CHCl_3_), and acetone (Fuyu Fine Chemical Co., Ltd.™, Tianjin, China) were purified through standardized techniques.

### 2.2. Synthesis of BHB

The synthetic pathway of BHB is shown in Figure 1a. HDI was added dropwise into BDO (molar ratio of BDO and HDI was 8:1) at 75 °C under mechanical stirring. Subsequently, the reaction proceeded for another 4 h at the same temperature. The system was cooled to ambient temperature, with the resultant mixture filtrated under vacuum. The crude product was washed several times using anhydrous acetone to render BHB as a white powder, which was consequently vacuum-dried to constant weight. The yield was 89.0%.

^13^C NMR (101 MHz, DMSO-d_6_): δ 156.5 (-NH*C*OO-), 63.6 (-*C*H_2_OOCNH-), 60.4 (-*C*H_2_OH), 40.1 (-*C*H_2_*C*H_2_CH_2_CH_2_CH_2_*C*H_2_-), 29.5(-CH_2_*C*H_2_CH_2_CH_2_*C*H_2_CH_2_-), 28.9 (-*C*H_2_CH_2_OH), 25.6 (-CH_2_CH_2_*C*H_2_CH_2_CH_2_CH_2_-), 25.3 (HOCH_2_CH_2_*C*H_2_-); MS (*m*/*z*) determined for C_16_H_32_O_6_N_2_ [M + Na^+^] 371.42 was 371.22.

### 2.3. Preparation of PEUs and PEU Films

A series of PEUs—with various chemical compositions—was prepared according to the formulations in Table 1, with a reaction overview illustrated in Figure 1b. Typical process: PCL and HDI were dissolved within anhydrous DMF (1.0 g/mL) at ambient temperature for obtaining a homogeneous solution. Post-purging with dry argon, Sn(Oct)_2_ (2 wt ‰ of PCL and HDI) was introduced and the reaction was conducted at 80 °C to prepare the prepolymer of NCO-terminated PCL (OCN-PCL-NCO). Once -NCO content (identified through di-n-butylamine-acetone titration [26]) reached the theoretical value range, DMF solution for BHB (0.2 g/mL) was introduced dropwise within the mixture, with such a reaction permitted to take place until absorption-peak for -NCO in FT-IR spectrum (2260~2275 cm^−1^) disappeared completely. Consequently, additional DMF was introduced to dilute the mixture. The diluted solution (0.05 g/mL) was de-aerated using diminished pressure and subsequently poured within a Teflon^®^ mold. The solvent was evaporated (40 °C) for obtaining PEU films (thickness: 0.3 ± 0.025 mm), which were termed PEU-X (X: molar ratio for HDI and PCL).

### 2.4. Characterization and Instruments

Characterization: NMR spectra were registered through the VANCE II^®^ 400 MHz NMR spectrometer (Bruker™, Rheinstetten, Germany). FT-IR spectra were collected within the Bruker™ Alpha^®^ spectrometer carrying attenuated total reflectance (ATR^®^, Bruker™) feature—at a frequency range of 4000–400 cm^−1^, with 4 cm^−1^ resolution. Mass spectrum was obtained on a Bruker™ maXis ultra HR-TOF-MS system with a methanol solution of ~10 µg/mL. Molecular weights (*M*_n_ and *M*_w_)/molecular weight distribution (*M*_n_/*M*_w_) were determined through GPC (Waters 2414^®^) with THF as the continuous phase and monodisperse polystyrene serving as control.

Thermal features: Thermo-gravimetric analysis (TGA) was conducted for establishing the thermal stability of polymer materials. This was executed with a thermo-gravimetric analyzer (TGA 2950^®^, TA Instrument™, New Brunswick, NJ, USA). Thermal weight loss behaviors were supervised, with a temperature range of 50–550 °C (heating rate = 10 °C/min under nitrogen (50 mL/min)). Differential-scanning calorimeter (DSC) was adopted for analyzing thermal transitions of the polymers using a DSC2910^®^ (TA Instrument™, USA) (−40 to 100 °C) under consistent nitrogen flow, while the heating rate was 10 °C/min. Individual samples were subjected to two back-to-back runs and registered, with such dataset outcomes obtained through a second heating iteration. 

Tensile features: Tensile behavior for polymer films were obtained on a universal testing machine (UTM100^®^, Shandong Wanchen Instrument™, Jinan, China). Based upon ISO527-2 protocol, dumbbell-shaped films (L: 4 mm, H: 30 mm) were analyzed through a crosshead speed set at 50 mm/min at ambient temperature.

Water absorption: The water-absorbing properties of such films was determined through the gravimetric method. A film sample (weighted as *M*_o_) was dipped into distilled water at 37 °C. At fixed time spans, the sample was removed, gently pressed between tissue papers, and weighed as *M*_w_. The measurement was performed until there was no increase in sample mass. The water absorption was obtained through the equation: Water absorption % = (*M*_w_ − *M*_o_)/*M*_o_ × 100.

Hydrolytic breakdown: Hydrolytic degradation for polymer films was determined gravimetrically within PBS solution at 37 °C. Film samples, which had reached the equilibrium for water absorption, were weighted (W_o_) and kept in PBS solution (pH = 7.4). Post-incubation for a predetermined time, samples were collected and weighted (W_d_). The percentage of remaining weight was ascertained according to the equation:Residual weight % = (W_o_ − W_d_)/W_o_ × 100.

All analyses were performed for 12 months or until the films became fragments. The typical breakdown sample was freeze-dried, and surface morphologies were observed through SEM (Hitachi™ SU8010^®^, Tokyo, Japan).

Cytotoxicity test: MTT methodology was employed for determining viability within L929 mouse fibroblast cultures and to evaluate PEU film cytotoxicity. Briefly, the film sample was rendered sterile through UV irradiation and consequently extracted in DMEM augmented through 10% FBS (37 °C/24 h). Cells (5 × 10^4^ cells/mL) were seeded into the filtered extract and incubated at 37 °C for 24, 48, and 72 h. Post-continuous culturing in MTT solution for another 5 h, dimethyl sulfoxide was introduced for dissolving any developed formazan pigment. Absorbance at 570 nm was determined through a microplate reader, and cell viability was obtained according to the control (cells incubated only within a culture medium).

## 3. Results and Discussion

### 3.1. Synthesis and Characterization 

BHB: The chain extender for aliphatic diurethane diol with ordered structure was generated from reacting HDI with a larger volume of BDO. The chemical structure of BHB was validated through ^1^H NMR, ^13^C NMR, and FT-IR. Within ^1^H NMR spectrum for BHB (Figure 2a), the peaks observed at δ 4.42, 3.90, 3.39, and 2.91 belonged to the proton signals of -OH, -COOCH_2_-, -C*H*_2_OH, and -C*H*_2_NH-, respectively. In addition, the core regional ratio for peaks was approximately 1(0.46):2(0.98):2(1.00):2(0.98), which was reflective of the BHB molecular frame (Figure 1a), suggesting that there was nil extended segment present within the product. Furthermore, characteristic absorption peaks of -N-H, amide I/II appeared at 3312, 1680, and 1537 cm^−1^ in Figure 3a, which demonstrated the formation of the variation of urethane groups. Furthermore, broad absorption bands at ~3565 cm^−1^ must be linked to terminated -OH for HBH. Such results in combination with ^13^C NMR and MS analyses indicated the successful synthesis of diurethane diol (BHB).

PEU: The PEUs, having varied OHS content, were generated through the classic two-step method. PCL diol was first reacted with HDI to prepare a prepolymer of OCN-PCL-NCO, which was consequently chain-extended with BHB to give PEUs. PEU films were collected through the solvent-evaporation technique. The number-average molecular weight (*M*_n_) and molecular weight distribution (*M*_w_/*M*_n_) for PEU materials were 75~89 KDa and 1.49~1.62 (Table 1), respectively. ^1^H NMR and FT-IR were employed for confirming PEU chemical structures (taking PEU-1.0 and -2.0 as representative samples). The signals from methylene protons associated with -OH within BHB (δ 3.39 ppm, Figure 2a) and PCL (δ 3.64 ppm, Figure 2b) vanished. Meanwhile, a novel signal in ^1^H NMR spectra of PEUs (Figure 2c,d) was observed at δ 3.15 ppm which belonged to the methylene protons which had linked onto formed carbamate groups. Regarding FT-IR spectra, absorption bands of -OH in BHB (~3530 cm^−1^, Figure 3a) and -NCO in OCN-PCL-NCO (~2275 cm^−1^, Figure 3c) disappeared in its entirety, indicating a positive chain extension. In addition, PEU-2.0 displayed a larger proton sign of -C*H*_2_-NH- (δ 4.76 ppm) within ^1^H NMR spectrum (Figure 2d) and stronger absorption intensity of urethane groups (N-H, amide I and II) in FT-IR spectrum (Figure 3e) than PEU-1.0 (Figure 2c and Figure 3d), which was attributed to the increased OHS content.

### 3.2. Thermal Stability

TGA was widely employed for analyzing the thermal degradation by materials at differing temperatures. Figure 4 illustrates the TGA curves of PEU films with various OHS content. Such curves describe that thermal degradation for the entirety of samples took place across two weight-loss phases. An initial phase at a lower temperature was mainly ascribed to the scission of ester bonds, while the second phase took place at an elevated temperature that corresponded to carbamate bonding breakdown [27]. In addition, PEU film thermal stability was promoted upon increasing OHS presence (PEU-1.0~PEU-2.0), which might be ascribed to the physical-linking networked structure developed through H-bonding not just across hard segments but also across hard/soft segments. Starting-weight reduction temperature for PEU films was higher than 200 °C, which meant that they could be high-temperature sterilized, although additional studies should be carried out.

### 3.3. Thermal Transition

DSC thermograms for PEU films with various OHS content are shown in Figure 5. PEU films represented a sole glass-transition temperature (*T*_g_), which indicated the homogenous nature of PEUs [28]. Furthermore, the *T*_g_ was altered gradually to an elevated temperature upon the increase of OHS content (PEU-1.0~PEU-2.0). The higher OHS content not only meant an increase in molecular rigidity but also produced more hydrogen bonds which formed the network structure. These reasons induced the restriction of polymer chain mobility, therefore no obvious microphase separation was found within thermograms and *T*_g_ was altered to elevated temperatures [29,30]. Thermograms also exposed a broad endothermic peak with melting temperatures (*T*_m_) at ~56 °C, which were distinctly due to the melting transition for crystallized PCL-soft/BHB-hard domains. Enthalpy for peaks decreased from 28.2 J/g to 15.9 J/g with an increase in OHS content (PEU-1.0~PEU-2.0). This should also be attributed to the denser H-bonds which hindered the rearrangement of soft and hard segments, leading to reduced crystallinity. A similar phenomenon had been found by Han in waterborne PU, based on PEG [31].

### 3.4. Tensile Features

Tensile property proves essential for the application of medical PU materials [32]. The representative stress-strain curves for PEU films having various OHS content are presented in Figure 6, with signature datasets of tensile strength and elongation at the break depicted in Table 2. No defined yielding point was denoted from curves, indicating an elastic deformation [33]. Upon increasing OHS content (PEU-1.0~PEU-2.0), elongation-at-break diminished, while tensile strength was enhanced. Similar tendencies had been found in the other PU materials. Dataset outcomes demonstrated tensile features can be regulated by modifying BHB dosage. In addition, all the PEU films showed good tensile features, especially for PEU-2.0 film, which contained the highest OHS content (BHB content: 20.7 wt %) and exhibited outstanding tensile features—having a tensile strength of 40.6 MPa and elongation-at-break of 477%. Such values were comparable to those of commercial medical PU, based on polycarbonate diol and 4,4′-diphenylmethane diisocyanate (Chronoflex AR^TM^, tensile strength: 41 MPa, elongation at break: 448%) [34]. The well-defined chain extender of BHB contained two carbamate groups, thus they would generate six carbamate groups in each hard segment containing BHB, which could form increased-density inter/intramolecular H-bonding and supply additional energy dissipation throughout tensile tests, resulting in outstanding tensile features.

### 3.5. Water Absorption

Regarding the breakdown of polyester-based polyurethane, this was predominantly occurring through ester bond hydrolysis, where water absorbency has an important effect on the degradation rate [35]. The water absorption of PEU films was tested as a function of immersion duration (Figure 7). All PEU films presented Fickian diffusion behavior [36,37] and reached the equilibrium level post-36 h. Although the equilibrium water absorption (EWA) was marginally raised upon OHS content reduction (PEU-2.0~PEU-1.0), all the films displayed a low EWA of less than 5%. This could be attributed to two factors: one is the hydrophobic chain structure; while another reason could be the compact H-bond-crosslinked network structure, which restricted water molecules from entering the matrix. 

### 3.6. Hydrolytic Degradability

The high biostability represents a paramount requirement to ensure long-term implant biomaterial quality. Hydrolytic degradability for PEU films was measured by the weight-loss method within PBS (pH 7.4) at body temperature, while weight-loss curves as a function of time are displayed in Figure 8. The weight-loss rate for films decreased drastically upon increased OHS level (PEU-1.0~2.0). The PEU-1.0 film, which did not contain OHS, had a weight loss of 41.2% post-7 months of degradation and became fragmented. While only 29.5% weight loss was detected for PEU-2.0 film (the highest OHS content across all samples) on assay termination, the film still maintained its primary shape and seemed to retain tensile features up to a certain degree. Undoubtedly, such a breakdown was predominantly induced through ester-bond hydrolysis within soft segments. When BHB was introduced into the main chain, six urethane groups contained in each OHS. The additional urethane groups could form denser H-bonds and produced a compact networked structure, which restricted water molecules from approaching ester groups, leading to a gradual hydrolysis rate. Such results were consistent with the water-absorbing capacity. The materials expressing slow biodegradability (or high biostability) can meet the requirement of long-term implants, although additional measurements need to be performed.

Weight loss for films during degradation-processing meant that several small molecules were formed and dissolved in water, thus the degradation process could be embodied by changes in surface morphology. The SEM images for PEU-2.0 films at differing degradation periods are displayed in Figure 9. A smooth and level surface was observed within the pristine film (Figure 9a). However, the relatively rough surface appeared post-degradation for 4- (Figure 9b) and 8-months (Figure 9c), the film also kept its starting shape and had no apparent structural change. Upon termination of measurements (12 months, Figure 9d), the surface exhibited distinct sunken morphology with considerable weight loss, though the films still maintained their integrity, demonstrating that the PEU-2.0 film possessed higher hydrolytic stability.

### 3.7. Cytotoxicity Assay

Aimed at biomedical applications, good biocompatibility is the most fundamental requirement, and consequently, cytotoxicity analysis is essential for assessing such a property. As shown in Figure 10, cell viability—as assessed through MTT assays—was slightly elevated upon increased OHS presence, indicating that the introduction of OHS contributed to cytocompatibility. This could be due to the compact networking structural formation through hydration bonds, which inhibited toxic substances from entering the culture medium. All PEU films containing OHS (PEU-1.25~2.0) exhibited cell viability > 75%, meaning that cytotoxicity was class 1 (ISO 10993.5-2009) and that the acceptable cytocompatibility could meet the requirement of implant biomaterials [38,39].

## 4. Conclusions

Within this study, ordered structure was introduced to the hard segments of PEUs by a facile and cost-effective method. The BHB, which contained an ordered structure, was synthesized and employed as a chain extender to react with NCO-terminated PCL to prepare the PEUs. The influences of OHS content on physicochemical features of the casted PEU films were investigated. Regarding each OHS containing six urethane groups, it was beneficial to form higher-density H-bonds and consequently produce a compact network structure, resulting in great tensile features, low water absorption, and a slow hydrolytic degradation rate for such PEU films. PEU-2.0 films, which possessed the highest OHS content, exhibited tensile features with a tensile strength of 40.6 MPa and elongation-at-break of 477%, equilibrium water absorption of 4.3 wt %, and weight-loss of 29.5% after post-12 months’ degradation. Furthermore, the cytotoxicity test for film extracts showed that cell viability for all PEUs containing OHS exceeded 75%, indicating a good cytocompatibility. Regarding the outstanding tensile features, high hydrolytic stability, non-toxic and absorbable degradation products, and acceptable cytocompatibility, these cost-effective materials (especially PEU-2.0) exhibited promising applications as long-term implants, although additional biological assessments are required.

## Figures and Tables

**Figure 1 polymers-14-01674-f001:**
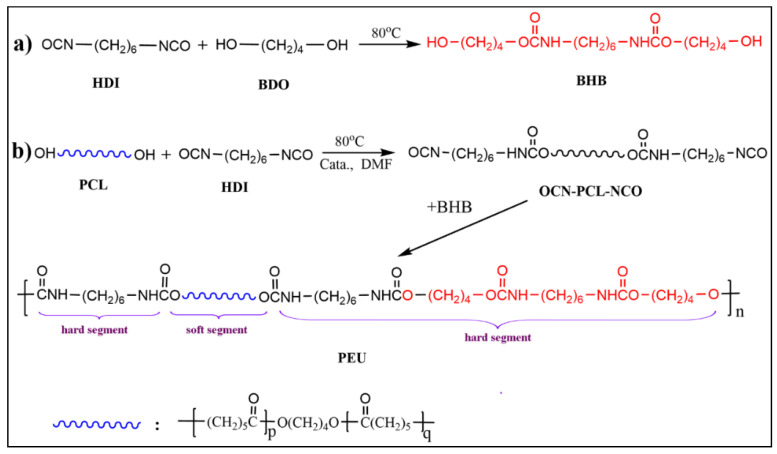
Synthetic-pathways of (**a**) BHB and (**b**) PEU.

**Figure 2 polymers-14-01674-f002:**
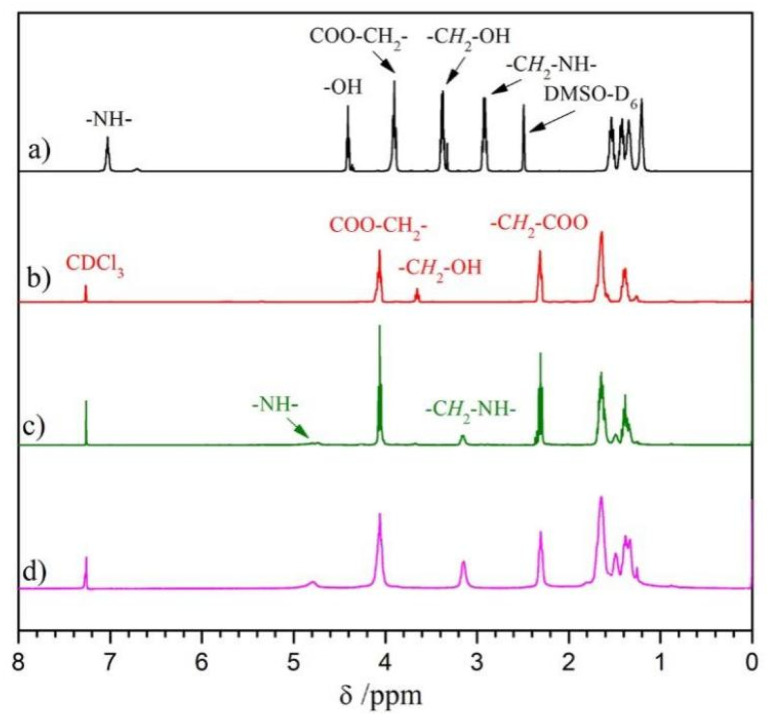
^1^H NMR spectra for (**a**) BHB, (**b**) PCL, (**c**) PEU-1.0 and (**d**) PEU-2.0.

**Figure 3 polymers-14-01674-f003:**
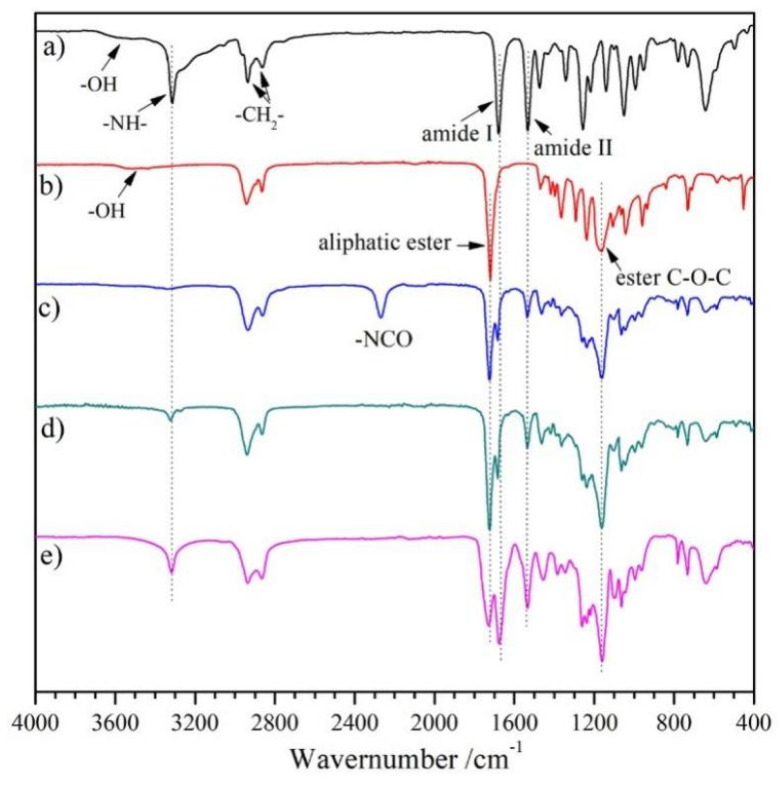
FT-IR spectra for (**a**) BHB, (**b**) PCL, (**c**) OCN-PCL-NCO, (**d**) PEU-1.0 and (**e**) PEU-2.0.

**Figure 4 polymers-14-01674-f004:**
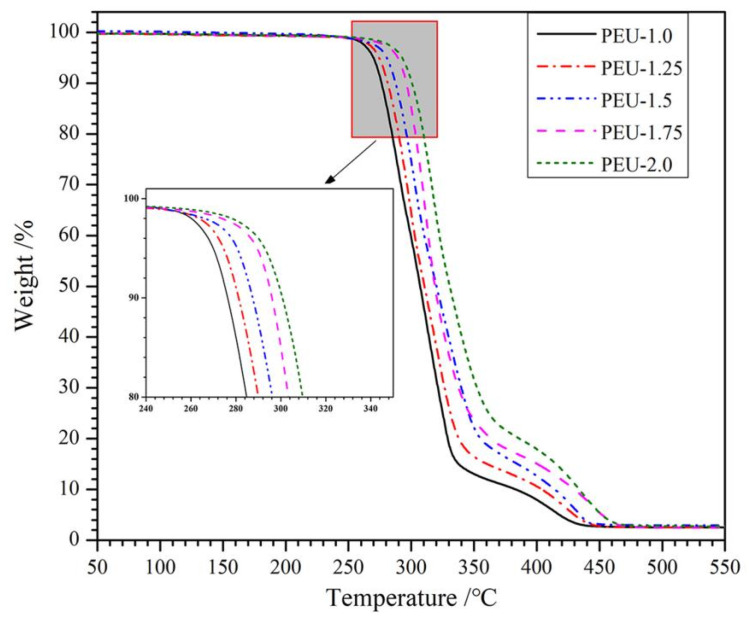
TGA curves for PEU films.

**Figure 5 polymers-14-01674-f005:**
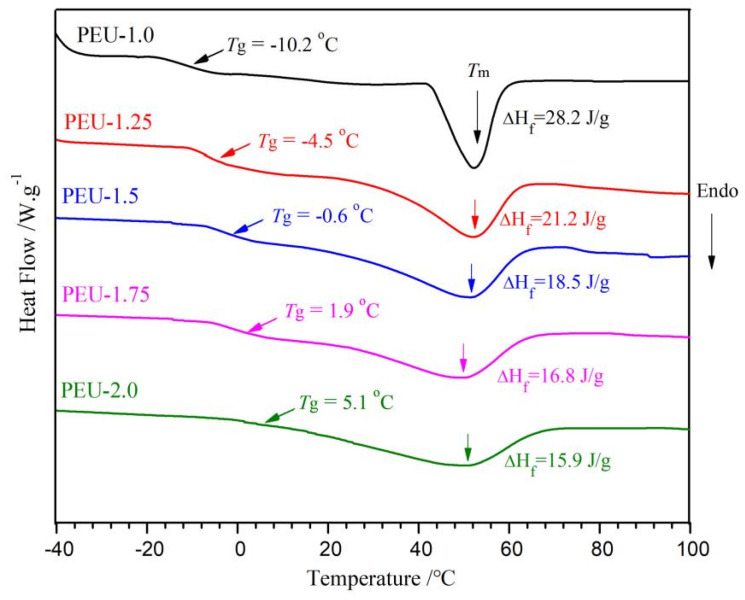
DSC thermograms regarding PEU films.

**Figure 6 polymers-14-01674-f006:**
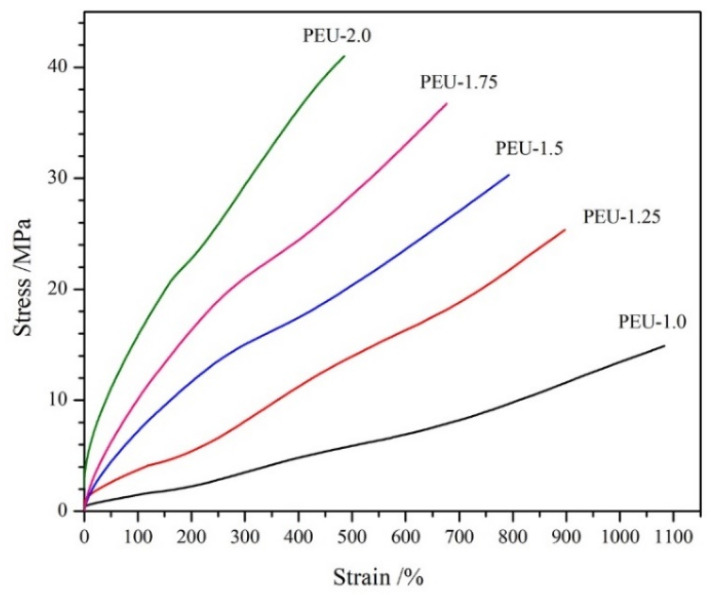
Representative stress-strain curves for PEU films.

**Figure 7 polymers-14-01674-f007:**
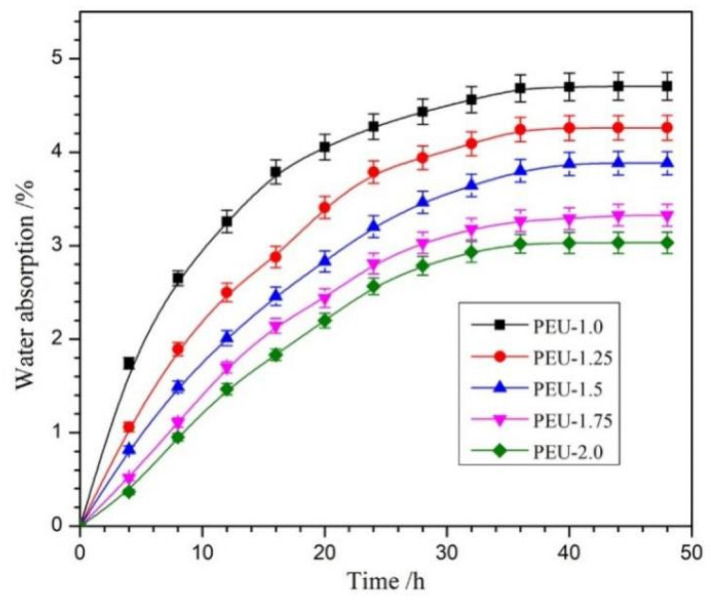
Water absorption of PEU films as function of immersion time (*n* = 3).

**Figure 8 polymers-14-01674-f008:**
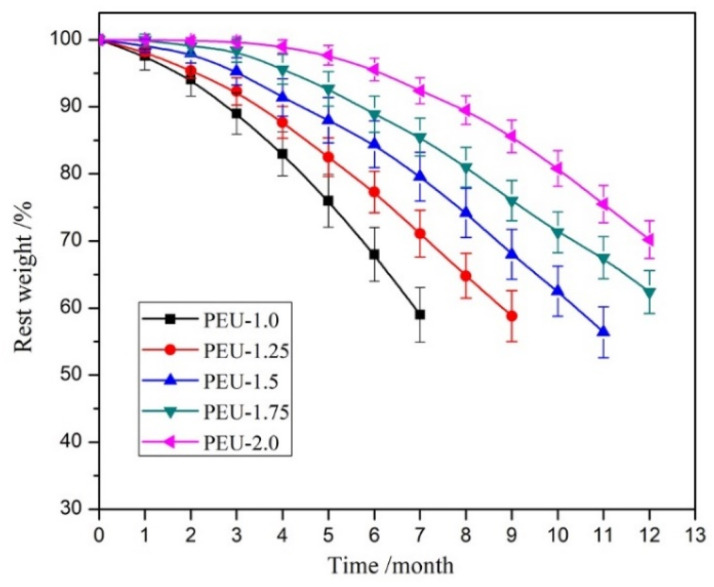
Weight-loss for PEU films as function of breakdown duration within PBS (pH: 7.4) at 37 °C (*n* = 5).

**Figure 9 polymers-14-01674-f009:**
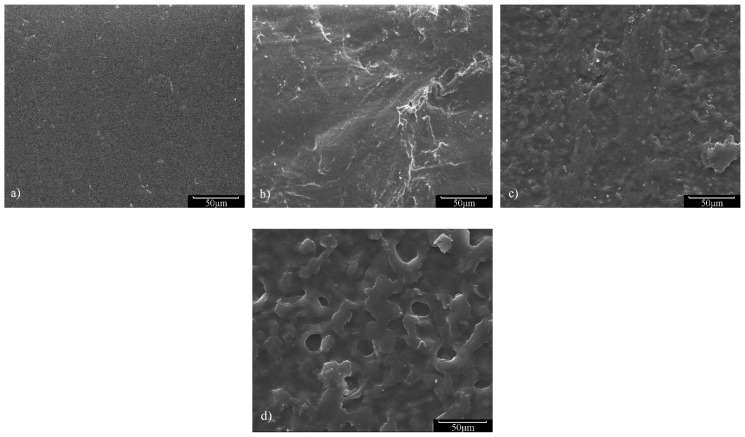
SEM images of PEU-2.0 film following (**a**) 0, (**b**) 4, (**c**) 8 and (**d**) 12 months’ degradation.

**Figure 10 polymers-14-01674-f010:**
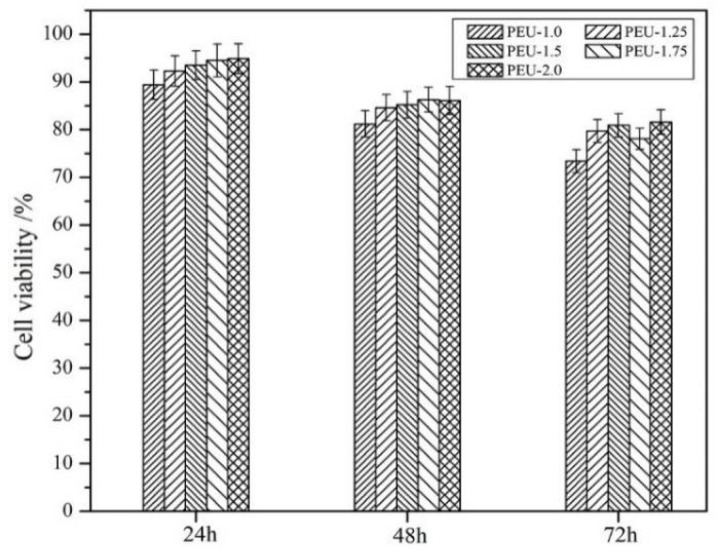
L929 cell viability in PEU film extracts (*n* = 3).

**Table 1 polymers-14-01674-t001:** PEU characteristics.

Samples	PCL /mmol	HDI /mmol	BHB	BHB Content /wt %	*M*_n_/KDa	*M*_w_/*M*_n_
PEU-1.0	5.0	5.0	0	0	89	1.49
PEU-1.25	5.0	6.25	1.25	6.7	85	1.51
PEU-1.5	5.0	7.5	2.5	12.2	78	1.62
PEU-1.75	5.0	8.75	3.75	16.8	76	1.58
PEU-2.0	5.0	10.0	5	20.7	75	1.53

**Table 2 polymers-14-01674-t002:** Tensile characteristic values for PEU films (*n* = 5).

Films	PEU-1.0	PEU-1.25	PEU-1.5	PEU-1.75	PEU-2.0
Tensile strength/MPa	15.1 ± 0.35	25.4 ± 0.62	30.6 ± 1.58	36.8 ± 2.42	40.6 ± 2.94
Elongation at break/%	1085 ± 78	903 ± 64	797 ± 56	678 ± 50	477 ± 41

## Data Availability

The data presented in this study are available on request from the corresponding author.

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
