# Peer review of "A Facile and Cost-Effective Method to Prepare Biodegradable Poly(ester urethane)s with Ordered Aliphatic Hard-Segments for Promising Medical Application as Long-Term Implants"

_polymers, 2022, doi:10.3390/polym14091674_

Round 1
Reviewer 1 Report
I have read the manuscript entitled “A Facile and cost-effective method to prepare biodegradable poly(ester urethane)s with ordered aliphatic hard-segments for promising medical application as long-term implants” and analyzed its potential for publication in the MDPI journal Polymers (ISSN 2073-4360).
In my opinion, the manuscript is interesting and covers a recent topic. The development of new biodegradable polymer is nowadays very soundly and important. After consideration I suggest accepting the manuscript for publication after minor revision.
Below I have listed most important weaknesses of the manuscript that I have spotted during my reading.
- Introduction is rather modest and in I have feeling that it does not build a solid foundation to justify the research objectives. I Suggest to extend the introduction in a way that build narration providing with the explanation of presented results scientific strength and novelty.
Reviewer 2 Report
The manuscript is focused on investigations of a biodegradable poly(ester urethane)s (PEUs) with ordered hard segments (OHS) designed for medical application as long-term implants.
The diurethane diol (1,4-butanediol-hexanediisocyanate-1,4-butanediol) (BHB) was firstly designed and synthesized to finally obtain PEUs. The chemical structures of both BHB and PEUs were investigated by NMR, ATR-FTIR. The OHS content contribution on physical/chemical features (thermal stability and transition, tensile features, water absorption, hydrolytic degradability) for casted PEU films was investigated. The cytotoxicity tests are presented.
This contribution gives new information, it is well written and original, well organized and explained. The topic is relevant to the field of Polymers journal.
Before publication, the authors have to suitably address to the following points:
i. It is claimed in the Abstract part that “MS together with GPC” will define the molecular structures for BHB and PEUs, but these investigation methods are neither introduced nor discussed within the manuscript.
ii. In case of the envisaged medical application, namely the long-term implants, the thickness control and adherence are of utmost importance. The casted method used to produce films is questionable from the uniformity and adhesion point of view. The given thickness of 0.3±0.25mm has a huge error level, while the risks of delamination are high (the film adherence is not approached at all).
iii. Another manuscript drawback consists of the absence of antimicrobial tests that are crucial in case of a log-term implants. Please introduce them within this manuscript.
